# Oral Supplementation of Ozonated Sunflower Oil Augments Plasma Antioxidant and Anti-Inflammatory Abilities with Enhancement of High-Density Lipoproteins Functionality in Rats

**DOI:** 10.3390/antiox13050529

**Published:** 2024-04-26

**Authors:** Kyung-Hyun Cho, Ji-Eun Kim, Myeong-Sung Lee, Ashutosh Bahuguna

**Affiliations:** Raydel Research Institute, Medical Innovation Complex, Daegu 41061, Republic of Korea

**Keywords:** apolipoprotein A-I (apoA-I), apoptosis, electron microscopy, fatty liver, glycation, high density lipoprotein, interleukin 6, oxidative stress, paraoxonase (PON)-1

## Abstract

Research on ozonated sunflower oil (OSO) is mostly restricted to its topical application, whereas the functional and toxicological assessment of oral OSO consumption is yet to be solved. Herein, OSO was orally supplemented in rats to assess the impact on plasma antioxidant status, low-density lipoproteins (LDL), and high-density lipoproteins (HDL). Also, the functionality of HDL from the OSO-supplemented rats (OSO-HDL) was tested against carboxymethyllysine (CML)- induced hyperinflammation in embryo and adult zebrafish. The results revealed that four weeks of OSO supplementation (3 g/kg BW/day) had no adverse effect on rats’ hematological and blood biochemical profiles. Nonetheless, decreased interleukin (IL)-6, and LDL-C levels, along with enhanced ferric ion reduction ability (FRA) and sulfhydryl content, were observed in the plasma of OSO-supplemented rats compared to the control and sunflower oil (SO) supplemented group. In addition, OSO supplementation stabilized apoA-I/HDL and augmented HDL-allied paraoxonase (PON)-1 activity. The microinjection of OSO-HDL (10 nL, 2 mg/mL) efficiently prevented the CML (500 ng)-induced zebrafish embryo mortality and developmental deformities. Similarly, OSO-HDL thwarted CML-posed neurotoxicity and demonstrated a significant hepatoprotective effect against CML-induced fatty liver changes, hepatic inflammation, oxidative stress, and apoptosis, as well as exhibiting a noticeable influence to revert CML-induced dyslipidemia. Conclusively, OSO supplementation demonstrated no toxic effects on rats, ameliorated plasma antioxidant status, and positively influenced HDL stability and functionality, leading to a protective effect against CML-induced toxicity in zebrafish.

## 1. Introduction

Ozone is a blue-colored gas with several medicinal properties that can be used to treat a variety of ailments. At a therapeutic dose, ozone substantially modulates nuclear factor kappa (NF*κ*B) and nuclear factor erythroid 2-related factor 2 (Nrf-2) activation and thus impacts the regulation of inflammatory [1] and antioxidant pathways [2,3]. Of note, many clinical studies have documented the efficient role of ozone therapy in maintaining lipid profile by reducing TC and TG levels in patients of hypertension [4], cardiopathy [5], and ischemic disease [4], along with the elevation of HDL-C in psoriasis patients [6]. Despite the multitude of benefits, the brief lifespan of ozone is consistently a cause of concern. Fortunately, this can be improved several-fold by ozone entrapment in a range of vegetable oils, including olive oil and sunflower oil (SO). Ozonated sunflower oil (OSO) is derived from the efflux of ozone to SO where ozone entrapped in the unsaturated site of the SO components through the Criegee mechanism leads to the production of a variety of carboxylic acids, aldehydes, ozonide, and peroxide species responsible for the diverse bio-functionality [7,8,9].

The effect of OSO has been well documented as an effective antimicrobial against a broad spectrum of microorganisms in vitro [10,11] and in animal studies [12,13,14]. A notable therapeutic effect of OSO to cure the human fungal diseases tinea pedis [15] and onychomycosis [14] has been observed. In addition, a remarkable effect of OSO in treating endometriosis in mares [12,16] and demodicosis in dogs [13] logged the broad-spectrum antimicrobial role of OSO. Besides the antimicrobial effect, we have recently disclosed the curative effect of OSO against atopic dermatitis (AD) in the mouse model [17], where we observed that the topical application of OSO significantly affected the infiltration of mast cells, filaggrin, and thymic stromal lymphopoietin (TSLP). In addition, OSO was logged for diminished serum Ig E, TNF-α, and IL-1β levels and displayed a significant antioxidant and anti-inflammatory effect by inhibiting the IL-4/STAT3/MAPK pathway and expression of NFκB [17].

Most of the investigation of OSO commenced on its topical application, and a limited study has documented the effective role of OSO intake by animals. However, in one of our recent studies, the health impact of OSO dietary consumption for two years in zebrafish has been documented [18]. The finding concluded that OSO consumption has a substantial antiaging effect that improves the functionality of the vital organs against age-associated stress manifested by antioxidant and anti-inflammatory effects [18]. Furthermore, in another study, we observed the effective function of OSO in curtailing high cholesterol diet-induced fatality and dyslipidemia in adult zebrafish [19]. The gastroprotective role of OSO through the activity enhancement of superoxide dismutase (SOD) and glutathione peroxidase (GSH-PX) in rats has suggested the therapeutic role of OSO [20]. Despite this limited number of studies, no comprehensive study has commenced on OSO intake precisely to evaluate OSO’s impact on plasma antioxidant activity, lipoprotein oxidation, and functionality of high-density lipoproteins (HDL).

In our preliminary study, we evaluated the positive impact of OSO on the activity of HDL-associated paraoxonase (PON)-1 and the structure and functionality of HDL [10]. However, the study was conducted based on the in vitro interaction of OSO with HDL and not on the in vivo effect of OSO consumption. To fill the gap, the present study was conducted to analyze the impact of OSO consumption over four weeks on the hematological, biochemical, and lipid profiles and the plasma antioxidant status of Sprague-Dawley rats, following the effect on LDL and HDL oxidation, glycation, and structural stability and functionality of HDL. In addition, the functionality of HDL was tested against CML-posed toxicity in embryos and adult zebrafish.

## 2. Materials and Methods

### 2.1. Materials

Ozonated sunflower oil (OSO) with characteristic Oleozon^®^ (783.4 mmol peroxide/kg, 2.42 mg potassium hydroxide/g acidity and 113.2 mPa.s viscosity) [14] was acquired from Raydel Pty., Ltd. (Thornleigh, NSW, Australia). Sunflower oil (SO) was purchased from the supermarket situated in Daegu. Unless otherwise stated, all the chemicals used were of analytical grade and used as supplied. A list of the used chemicals is provided as Appendix A.

### 2.2. Rat Maintenance, Experimental Design, and Hematological and Biochemical Analysis

Adult Sprague-Dawley rats [6 weeks aged, ~150 g body weight (BW)] were housed in a pathogen-free environment at 24 ± 2 °C, 55 ± 2% relative humidity, and a periodic light and dark cycle of 12 h adhering to the guidelines of the Animal Ethical Committee Daegu Catholic University (approval number IACUC-2023-027, 8 August 2023, Daegu, Republic of Korea). The animals were fed on the normal pellet diet and water ab libitum. After two weeks of acclimatization in the conditions mentioned above, rats (*n* = 30) were assorted randomly into three separate groups (*n* = 10/group) and supplemented with a pre-optimized dose of 3 g/kg BW/day (Appendix A) of SO or OSO or distilled water following the oral gavage as a previously described method [21]. The distilled water 3 g/kg BW/day was orally administered using gavage in the control group. At the same time, 3 g SO or OSO/kg BW/day was orally administered by gavage in the SO and OSO-supplemented groups. The treatment in three groups continued for 4 weeks. Before the treatment, the BW of the rats across all the groups was measured to facilitate a similar amount of treatment (i.e., 3 g/kg BW/day). The equivalent dose was prepared in 1.5 mL tubes and administered orally via gavage. The respective tubes were washed with 0.25 mL water, and the suspended content was readministered orally. A similar procedure was followed for each rat in a different group to ensure nearly complete and uniform dose administration. Finally, after 4 weeks of treatment, the animals were sacrificed, and blood was immediately collected in ethylenediaminetetraacetic acid (EDTA)-containing tube.

Blood (1 mL) collected from the tail vein across the different groups was centrifuged at 4000× *g* for 15 min, and the collected plasma was processed for the hematological, biochemical, and blood lipid analysis using ADVIA 2120 hematology autoanalyzer (Forchheim, Bavaria, Germany) and Konelab Arena 20XT clinical chemistry analyzer (Waltham, MA, USA).

### 2.3. Ferric Ion Reduction Ability and Sulfhydryl Group Estimation

Ferric ion reduction ability (FRA) was determined by a previously described method with modification [22]. In brief, 20 μL plasma was mixed with 180 μL FRA reagent (prepared by blending 10 mL of acetate buffer (0.2 M, pH 3.6) with 1.25 mL each of 2,4,6-tripridyl-S triazin (10 mM) and FeCl_3_ (20 mM). FRA reagent devoid of plasma was also prepared, which served as blank. After 20 min incubation at room temperature, the FRA was determined by taking absorbance at 593 nm.

The sulfhydryl group was quantified by mixing 60 μL of plasma (1 mg/mL protein) with 60 μL of 5,5′-dithio-bis-(2-nitrobenzoic acid) (DTNB) (4 mg/mL). After 12 h incubation at room temperature, absorbance at 412 nm was determined, and sulfhydryl groups were quantified utilizing 13,600 M^−1^ cm^−1^ molar extinction coefficient (ε) of DTNB.

### 2.4. Isolation of Lipoproteins and Protein Quantification

Low-density lipoproteins (LDL) and high-density lipoproteins (HDL) from the blood of rats were extracted by the sequential density gradient ultracentrifugation method [23]. In brief, 6 mL of pooled blood underwent centrifugation at 4000× *g* for 20 min to yield plasma. A 2.5 mL aliquot of plasma was mixed with NaCl and NaBr solution maintained at density 1.019 < d < 1.063 and 1.063 < d < 1.225, respectively, followed by ultracentrifugation at 100,000× *g.* After 22 h ultracentrifugation, the LDL (1.019 < d < 1.063) and HDL (1.063 < d < 1.225) were collected from their respective density zones and processed for overnight dialysis against Tris-buffered saline (pH 8.0). The dialyzed LDL and HDL were preserved at low temperatures for further use. Protein quantification in the LDL and HDL was assessed using the Lowry method with slight modification following the earlier adopted method [24].

### 2.5. Evaluation of Glycation and Oxidation Extent

The extent of glycation in LDL and HDL was examined using the fluorescent spectroscopic method described previously [23]. In brief, the LDL or HDL was mixed in 500 μL of 0.2 M potassium phosphate-3 mM sodium azide buffer (pH 7.4) to achieve the final concentration of 2 mg/mL of LDL or HDL. Finally, the glycation extent was determined by measuring the fluorescent intensity (FI) at 370 nm (emission wavelength) and 440 nm (excitation wavelength).

The extent of oxidation in LDL and HDL was examined by a thiobarbituric acid reactive substance (TBARS) assay using malondialdehyde (MDA) as a standard following the method described earlier [25]. In brief, HDL or LDL (1 mg/mL, 50 μL) was mixed with 50 μL of 20% trichloroacetic acid (TCA), followed by the addition of 100 μL of 0.67% thiobarbituric acid (TBA). After 10 min incubation at 95 °C, an absorbance of 560 nm was recorded.

### 2.6. Morphological Analysis of LDL and HDL

The morphology of the LDL and HDL was examined by the transmission electron microscopic (TEM) analysis following the method described earlier [26]. In brief, an equal volume at LDL or HDL (1 mg/mL) was blended with phosphotungstate (2%, pH 7.4) and the suspension (5 μL) was spread over a 200-mesh carbon-coated copper grid. After 2 min incubation at room temperature, the excess of the suspension was blotted, and the grid was incubated at 50 °C. After 2 h incubation, the sample was visualized at 150*K* magnification under TEM (HT-7800, Hitachi, Tokyo, Japan) at 80 kV acceleration voltage.

### 2.7. Paraoxonase (PON)-1 Assay

The paraoxonase (PON)-1 activity of the isolated HDLs was determined by following the earlier described method [26]. In brief, 20 μL of HDL (2 mg/mL) was mixed with 180 μL of buffer [Tris-HCl (90 mM), NaCl (3.6 mM), CaCl_2_ (90 mM)] containing the paraoxon-ethyl substrate (0.55 M). After 60 min incubation at 25 °C, an absorbance (415 nm) was recorded to quantify the production of *p*-nitrophenol, a hydrolysis product of paraoxon-ethyl. Results are expressed as μU/L/min employing the molar absorbance coefficient 17,000 M^−1^cm^−1^ for *p*-nitrophenol.

### 2.8. Zebrafish Husbandry and Embryo Collection

Zebrafish were maintained at 28 °C (water temperature) with a photoperiod of 14 h (light) and 10 h (dark) following the guidelines of the Committee of Animal Care and Use of Raydel Research Institute, Daegu, Republic of Korea (approval code RRI-20-003). Zebrafish were fed on the normal diet (Tetrabit Gmhb D49304, Melle, Germany). To obtain embryos, male and female zebrafish were kept in the breeding tank and separated from each other using a physical divider. After 16 h segregation, the divider was removed, and male and female zebrafish were allowed to mate (~30 min) in the dark. The produced embryos were gathered, rinsed with tap water, and used for subsequent experiments.

### 2.9. Effect of Rat Plasma on Survivability of Zebrafish Embryos

The zebrafish embryos at 1.5 h post fertilization (hpf) were randomly distributed into five groups (*n* = 135/group). Ten nL of phosphate-buffered saline (PBS) was injected in the control-PBS group, while 500 ng carboxymethyllysine (CML) suspended in 10 nL PBS was injected in the CML group. The embryos in the DW, SO, and OSO groups were injected with 500 ng CML suspended in 10 nL of plasma obtained from DW, SO, and OSO-fed rats, respectively. The embryos in the different groups were periodically (5–72 h post-treatment) monitored to examine embryo survivability and developmental defects by accessing embryo coagulation, hatching, and somites number adhering to the outline principles of OECD 2019 [27].

### 2.10. Determination of Reactive Oxygen Species (ROS) and Apoptosis in Zebrafish Embryos

Reactive oxygen species (ROS) production in the embryos was examined by dihydroethidium (DHE) fluorescent staining following the earlier defined procedure [19]. In brief, zebrafish embryos (5 h post-treatment, *n* = 10) from each group were suspended in 0.5 mL of DHE (30 μM) for 30 min in the dark. Subsequently, stained embryos were rinsed with 1 × PBS and visualized under fluorescent microscopy (excitation = 585 nm and emission = 615 nm). Acridine orange (AO) fluorescent staining following the previously described method [19] was employed to evaluate the extent of apoptosis in the embryos. In brief, 10 embryos (5 h post-treatment) from distinct groups were stained with 0.5 mL of AO (5 μg/mL). After 30 min incubation, fluorescent images were captured under fluorescent microscopy (excitation = 505 nm, emission = 535 nm). The fluorescent intensity corresponding to DHE and AO-stained intensity was quantified by Image J software (http://rsb.info.nih.gov/ij, 16 January 2023 assessed version 1.53r).

### 2.11. Effect of Rat HDL on Survivability of Zebrafish Embryos

Zebrafish embryos were randomly divided into 5 groups (*n* = 80/group). Ten nL PBS was injected in the control PBS group. The CML group was injected with 500 ng CML/10 nL of PBS. The DW-HDL, SO-HDL, and OSO-HDL groups were injected with 500 ng CML dissolved in 10 nL of HDL solution (2 mg/mL) obtained from the rat plasma fed with DW, SO, and OSO, respectively. The survivability, developmental deformities, and fluorescent imaging (DHE, AO) were assessed across all the groups following the method mentioned in Section 2.9 and Section 2.10.

### 2.12. Effect of Rat HDL on Carboxymethyllysine Induced Toxicity in Adult Zebrafish

CML 250 μg (equivalent to a 3 mM amount considering the ~300 mg average body weight of zebrafish) was intraperitoneally injected to induce acute inflammation in adult zebrafish [28]. The 16-week-old zebrafish were randomly allotted to 5 groups (*n* = 20/group). Zebrafish in the PBS control group received an intraperitoneal injection of 10 μL PBS, while the zebrafish in the CML group were intraperitoneally injected with 250 μg CML/10 μL PBS. Zebrafish in the DW-HDL, SO-HDL, and OSO-HDL groups received intraperitoneal injections of 250 μg CML dissolved in 10 μL HDL (1 mg/mL) isolated from rat plasma fed with DW, SO, and OSO, respectively. Across all the groups, intraperitoneal injection was made using a 28-gauge needle after anesthetizing the zebrafish by submerging them in 2-phenoxyethanol (final 0.1%). The swimming activity of zebrafish was monitored at 30 min and 60 min post-treatment, while the survivability was monitored up to 3 h post-treatment following the OECD 2019 guidelines [27]. Zebrafish (*n* = 5/group) were sacrificed (at 3 h post-treatment) using hypothermic shock [19], and immediately, blood was collected in the tubes containing 1 mM EDTA-PBS (3 μL). The liver was surgically excised and preserved at −70 °C for further use.

### 2.13. Liver Histology and Immunohistochemistry (IHC)

The hepatic tissue was fixed with 10% formaldehyde and dehydrated with alcohol and subsequently embedded in paraffin. Hepatic tissue sections (5 μm thick) were obtained by microtome and processed for hematoxylin and eosin (H&E) staining [19] to examine the morphological variations. For oil red O (ORO) staining, a 5 μm thick tissue section was flooded with ORO stain [19]. After 5 min incubation at 60 °C, the tissue section was rinsed with 60% isopropanol and subsequently stained with hematoxylin for 30 s. Eventually, the stained tissue section underwent a water rinse before being observed under a microscope.

The 5 μm thick hepatic tissue sections were covered with 200× diluted primary antibody specific to interleukin (IL)-6 (ab9324, Abcam, London, UK). After 18 h incubation at 4 °C, the section was processed using an EnVison + system-HRP polymer kit (Code K4001, Dako, Glostrup, Denmark) with 1000× diluted HRP-conjugated secondary antibody.

DHE and AO fluorescent staining was performed by applying 250 μL of DHE (30 mM) and AO (5 μg/mL), respectively, on the 5 μm thick tissue sections. After 30 min incubation in the dark, the tissue section was visualized under the fluorescent microscope at 585 nm (excitation)/615 nm (emission) and 505 nm (excitation)/535 nm (emission) for the detection of DHE and AO fluorescent stained areas, respectively.

### 2.14. Blood Lipid Profile and Hepatic Function Biomarker Analysis

Triglycerides (TG) and total cholesterol (TC) were assessed using a commercial kit (TGs, Cleantech TS-S, and cholesterol, T-CHO; Wako Pure Chemical, Osaka, Japan), adhering to the recommended method provided by the manufacturers. The HDL-C, aspartate transaminase (AST), and alanine transaminase (ALT) were determined using commercial kits (AM-202, AM-103K, AM102K; Asan Pharmaceutical, Hwasung, Republic of Korea), following the manufacturers guidelines.

### 2.15. Statistical Analysis

The results are presented as mean ± SD (standard deviation) of three separate experiments. Statistical distinctions between the groups were determined by using one-way analysis of variance (ANOVA) following Dunnett’s test for pairwise comparison and Tukey’s post hoc analysis for multiple group comparison, utilizing Statistical Package for the Social Sciences software program (version 23.0; SPSS, Inc. Chicago, IL, USA).

## 3. Results

### 3.1. OSO Supplementation Posed Nontoxic Effects in Rats

A 4-week supplementation of SO and OSO (3 g/kg BW/day) exhibited no indication of toxicity in rats. No mortality of rats was observed in the SO- and OSO-supplemented groups. As compared to the initial day (week 0) BW, a significant ~1.9-fold (*p* < 0.001) enhancement in the BW was observed after 4-week supplementation of DW, SO, and OSO (Table 1). However, the BW enhancement between the groups remained non-significant, irrespective of the supplementation of DW, SO, or OSO (Table 1).

Consistent with the BW, no change in the hematological profile was observed in the SO and OSO-supplemented group, which was identified as the hematological profile of the control (DW) group (Table 2). However, slightly reduced eosinophil counts were observed in the OSO-supplemented group compared with the SO-supplemented group.

Most of the blood biochemical profiles among the groups are statistically similar and are within the reference range (Table 3). Interestingly, significantly higher albumin was quantified in the OSO-supplemented group compared to the DW and SO-supplemented groups. Also, a significant 33.3% reduced IL-6 level was detected in the OSO-supplemented group compared to the SO-supplemented group (Table 3).

The blood lipid profiles concerning TC and TG remain nearly similar across the groups, i.e., within the reference range. However, a decreased LDL-C level was observed in the OSO-supplemented group, which was significantly 33.2% (*p* < 0.032) and 36.5% (*p* < 0.017) lower than the LDL-C level observed in the DW (control) and SO-supplemented groups, respectively. Also, a significantly 50% lower LDL-C/HDL-C level was quantified in the OSO-supplemented group compared with the SO-supplemented group (Table 3). In addition, the histological outcome (H&E staining) of the liver, kidney, and heart revealed no toxicological effect of OSO (Appendix A). The combined results suggest that OSO supplementation has no toxic effect; moreover, OSO supplementation substantially alleviates IL-6 levels and LDL cholesterol.

### 3.2. OSO Supplementation Improved the Antioxidant Status

The effect of SO and OSO supplementation on the antioxidant status of the rat was examined by FRA assay (Figure 1A). Results revealed a substantial impact of OSO on serum ferric ion reduction compared to the SO and DW (control) groups. The rats supplemented with OSO displayed a significantly 1.3 and 1.5-fold (*p* < 0.001) higher FRA value than SO- and DW-control groups, respectively. Unlike OSO, the SO-supplemented group displayed non-significant changes in FRA compared to the DW (control) group.

In addition to higher FRA, the OSO-supplemented group displayed a higher plasma sulfhydryl content, which is significantly 2.1-fold, and 2.5-fold (*p* < 0.001) higher than the sulfhydryl-content measured in SO- and DW-consuming groups (Figure 1B). Combined results based on FRA and sulfhydryl content showed the substantial effect of OSO supplementation on improving plasma antioxidant status.

### 3.3. Plasma from OSO Protects Zebrafish Embryos against CML-Toxicity

A comparative effect of the plasma obtained from SO- and OSO-supplemented groups to protect zebrafish embryos against the toxicity posed by CML is depicted in Figure 2. The injection of CML (500 ng) displayed a severe embryo mortality, as evidenced by only 15.3% embryo survivability at 24 h post-treatment. Contrary to this, the highest embryo survivability was observed in the PBS-injected group (93.7%) at 24 h post-treatment. CML-induced embryo mortality was significantly prevented by inserting the rat plasma obtained across all the groups. However, the most promising effect on embryo survivability (59.4%) was displayed by the injection of the plasma obtained from the OSO-supplemented group, followed by DW (33.6%) and SO (21%) supplemented groups. As compared to the only CML-injected group, a significantly 3.9-fold (*p* < 0.001), 2.2-fold (*p* < 0.01), and 1.4-fold (*p* < 0.05) better embryos survivability was observed in the OSO, DW, and SO injected groups, respectively after 24 h post-treatment.

The developmental changes in the embryos among the different groups were measured at 24 h and 72 h post-treatment. Substantial developmental deformities and a significant reduction in somite counts were observed in most of the surviving embryos in the CML-injected group. As depicted in Figure 2B, severe tail fin curvature and pericardial edema were observed in the CML-injected group. Similarly, the bulk of surviving embryos that received CML co-injected with plasma obtained from DW and SO-fed rats displayed impairment of eyes and tail fin development and lower somite counts. Unlike this, the majority of embryos (~86%) injected with plasma obtained from the OSO-fed group efficiently prevented the CML-posed developmental defects and restored the somite counts.

The fluorescent staining of DHE and AO, as shown in Figure 2C,D, illustrates the generation of ROS and the occurrence of apoptosis in zebrafish embryos. The higher DHE fluorescent intensity in the CML injected group implies massive ROS production in response to CML injection. In contrast, in the PBS-injected group, the least fluorescent intensity was detected, representing the basal cellular level of ROS. A notable 1.4-fold (*p* < 0.05) and 1.9-fold (*p* < 0.001) reduced DHE fluorescent intensity was observed in DW- and OSO-treated groups compared to the group injected with CML, testifying to the efficacy of plasma from DW and OSO groups in suppressing CML-induced ROS generation. Moreover, compared to the DW group, the OSO-supplemented group exhibited a significantly 1.4-fold lower DHE fluorescent intensity, underscoring the heightened effectiveness of plasma derived from the OSO group in ROS production inhibition.

Surprisingly, the plasma obtained from the SO group showed no protective effect against CML-induced ROS production.

Analogous to the findings of DHE staining, the highest AO fluorescent intensity corresponding to apoptosis was observed in the CML-injected group (Figure 2C,E). The injection of the plasma from DW- and OSO-supplemented rats showed a significant 1.4-fold (*p* < 0.05) and 2.3-fold (*p* < 0.01) reduced fluorescent intensity compared to the CML injected group, indicating the influential effect of DW and OSO plasma to curtail CML posed apoptosis. While compared to the DW group, the OSO group showed a 1.8-fold decline in AO fluorescent intensity, implying the better efficacy of OSO in improving plasma activity against CML-induced apoptosis. The plasma obtained from the SO group was found ineffective in countering CML-induced apoptosis. The findings indicate that plasma mainly obtained from the OSO-supplemented group prevents ROS generation and apoptosis, consequently enhancing embryo survival rates and mitigating developmental abnormalities caused by CML.

### 3.4. Impact of OSO on the Attribute of LDL and HDL

The effect of SO and OSO consumption for 4 weeks on rat LDL and HDL composition and properties was determined (Table 4). Results revealed non-significant LDL composition changes concerning TC and TG constituents among the groups. Similarly, no changes in MDA levels were observed across the groups, signifying no adverse effect of SO and OSO consumption on LDL lipid peroxidation. However, a significant positive impact of OSO supplementation to prevent LDL glycation was perceived as apparent by 5.7% and 7.9% lower FI (glycated) compared to SO- and DW-supplemented groups, respectively.

Corresponding to the findings of LDL, the TC and TG composition of HDL remain almost similar across all the groups. However, a considerable lipid peroxidation preventive effect of OSO was noticed, evident by a 1.5-fold and 1.2-fold lower MDA level in HDL compared to the SO- and DW-supplemented groups, respectively. Also, the HDL extracted from the OSO-consuming group showed minimum glycation (FI), which is significantly 9.7% lower (*p* < 0.002) than the DW-supplemented (control) group. These outcomes clearly show that OSO composition substantially affects the inhibition of LDL and HDL glycation.

### 3.5. Transmission Electron Microscopic (TEM) Imaging of LDL and HDL

The TEM analysis displayed a nearly similar LDL particle morphology (circular) with an average particle diameter of ~21 nm and an average particle size of ~370 nm^2^ among all the groups (Table 4 and Figure 3). Contrary to the LDL, the size of HDL particles in the OSO-supplemented group (260.1 nm^2^) significantly increased by 30.3% (*p* < 0.001) and 55.8% (*p* < 0.001) in comparison with the particle size observed in DW (199.6 ± 10.6 nm^2^) and SO (166.9 ± 7.1 nm^2^) supplemented groups, respectively. These results demonstrate the modulatory effect of OSO consumption on the HDL particle size.

### 3.6. Enhanced apoA-I Stability and Paraoxonase (PON)-1 Activity in HDL in OSO Group

As shown in Figure 4A, a 0.23 nm red shift in WMF (337.62 nm) of apoA-I from the SO-supplemented group was noted compared to the DW-supplemented group WMF (337.62 nm), suggesting that SO group showed slightly more exposure of intrinsic Trp toward the polar phase. In contrast, a significant 0.5 nm blue shift (*p* < 0.01) in WMF (337.62 nm) of apoA-I from the OSO-consuming group was observed compared to the DW-supplemented group, suggesting that the OSO group might have the more distinct band and stabilized apoA-I in HDL particle via movement of intrinsic Trp toward the nonpolar phase.

In the same context, a comparative PON-1 activity observed in regarding HDL functions among the three groups is depicted in Figure 4B. The HDL from the OSO-supplemented group (OSO-HDL) displayed a significantly 1.6-fold and 1.7-fold (*p* < 0.01) enhanced PON-1 activity (50.1 ± 2.3 μU/L/min) as compared to HDL from DW group (31.4 ± 1.3 μU/L/min) and SO group (28.6 ± 0.5 μU/L/min), respectively. The outcomes illustrate the affirmative role of OSO supplementation in improving the functionality of HDL by enhancing the PON-1 activity. Taken together, the higher expression of apoA-I and increased blue-shift of WMF in the OSO group shows that consumption of OSO has a beneficial effect on HDL structure and functions.

### 3.7. HDL from OSO-Fed Rats Attenuated CML-Induced Embryo Toxicity

The previous results demonstrated the modulatory effect of OSO supplementation on HDL quality; furthermore, the functionality of the HDL isolated from OSO groups was tested using zebrafish embryos and compared with HDL obtained from DW and SO groups. Results depicted in Figure 5A displayed a severe toxic effect of CML exposure on the survivability of the zebrafish embryos, which was significantly improved by treatment with HDLs. Among the HDL treatments, the highest embryo protective effect was displayed by the OSO-HDL, followed by DW-HDL and SO-HDL. The embryo survivability in the OSO-HDL, DW-HDL, and SO-HDL treated groups was 75.5%, 45%, and 35%, respectively, which is significantly 3.4-fold (*p* < 0.001), 2.1-fold (*p* < 0.01), and 1.6-fold (*p* < 0.05) higher than the survivability of embryos in CML group (11%) at 24 h post-treatment. The OSO-HDL group displayed a 1.7-times and 2.2-times higher embryo survivability compared to the DW-HDL and SO-HDL treated groups, signifying the higher functionality against CML-posed mortality of zebrafish embryos.

The morphological changes in developing embryos were examined at 24 h and 72 h post-treatment, revealing significant developmental deformities in the embryos of the CML group (Figure 5B). Most of the surviving embryos (95%) in the CML-injected group appeared to have stunted growth, severe tail fin curvature, pericardial edema, and the lowest somite counts (19). Likewise, most of the surviving embryos in the SO-HDL group displayed stunted growth, tail fin curvature, pericardial edema, and reduced somite count (24). Compared to SO-HDL, the DW-HDL treatment effectively reverted CML-posed developmental defects. However, a mild tail fin curvature and pericardial edema were noticed in the ~40% of the surviving embryos. Contrary to this, the embryos treated with OSO-HDL showed the most profound effect against CML-induced developmental deformities. Most of the surviving embryos (~93%) in the OSO-HDL group displayed normal developmental morphology with improved somite counts (35), which is analogous to the zebrafish embryo development that appeared in the PBS-injected group.

The DHE and AO fluorescent imaging, as depicted in Figure 5C–E, showed the ROS level and apoptosis in the zebrafish embryos treated with CML or CML together with different HDLs. A massive ROS production was noticed in response to CML injection, which was significantly 1.6-fold (*p* < 0.05) and 2.2-fold (*p* < 0.01) curtailed by the treatment of DW-HDL and OSO-HDL, respectively. Compared to DW-HDL, the OSO-HDL treated embryos showed a 1.5-fold lower ROS level, attesting to the higher efficacy of OSO-HDL in inhibiting CML-induced ROS production. Importantly, no effect of SO-HDL was observed on the inhibition of CML-induced ROS production.

The AO staining showed a significantly 1.8-fold (*p* < 0.05) reduced AO fluorescent intensity in the OSO-HDL treated group as compared to the CML treated group, suggesting the influential role of OSO-HDL in inhibiting the CML-induced apoptosis in zebrafish embryos. Contrary to this, DW and OSO-HDL treatments were ineffective in countering CML-induced apoptosis.

### 3.8. Effect of HDL from OSO Fed Rats on CML-Induced Acute Paralysis and Mortality of Adult Zebrafish

As depicted in Figure 6A, high mortality with 43.3% survivability was observed in the CML-injected group against 100% survivability in the PBS-injected group after 3 h post-treatment, which signifies the severe toxic effects of CML. The injection of DW-HDL, SO-HDL, and OSO-HDL effectively prevents CML-induced zebrafish mortality. The OSO-HDL injected group displayed the utmost protective effect against CML-provoked mortality, as evidenced by the highest zebrafish survivability (80%), followed by SO-HDL (66.7%) and DW-HDL (56.7%), which is significantly1.8-fold (*p* < 0.001), 1.5-fold (*p* < 0.01), and 1.2-fold (*p* < 0.05) higher than the zebrafish survivability observed in the CML injected group. A 1.4-times and 1.2-times higher zebrafish survivability was observed in the OSO-HDL group than in the DW-HDL and SO-HDL injected group.

At 30 min post-injection, severe paralysis was observed in the CML-injected group, where all the fish were lying at the tank’s bottom, showing no sign of swimming (Figure 6B,C). The CML co-injected with DW-HDL or SO-HDL also displayed no sign of recovery against the CML-posed paralysis at 30 min post-injection. In contrast, OSO-HDL injection efficiently rescued the zebrafish from CML-induced paralysis (Appendix A). At 30 min post-treatment with OSO-HDL, zebrafish swimming activity showed a 30% improvement that progressively increased and attained the maximum of 70% at 60 min post-treatment, i.e., significantly 3-times (*p* < 0.001) and 4.4-times (*p* < 0.001) better than the swimming activity observed in the CML injected group at the respective time points. Unlike the 30 min observations, the DW-HDL and SO-HDL injected groups also showed significant restoration of CML-impaired swimming activity at 60 min post-treatment. At 60 min post-treatment, a significantly 3.2-fold (*p* < 0.001) and 3-fold (*p* < 0.001) higher restoration of CML-impaired swimming activity was observed in the DW-HDL and SO-HDL injected groups than in the CML injected group. The results showed that the HDL, mainly OSO-HDL, has a substantial curative effect on countering CML-induced mortality and paralysis in adult zebrafish.

### 3.9. OSO from Rats Prevents CML-Induced Hepatic Damage

The H&E staining in Figure 7A,B showed hepatic degeneration and higher neutrophil infiltration (indicated by red arrows) in the livers of the CML injected group. Conversely, treatment with OSO-HDL substantially prevents the CML-induced hepatic damage apparent by a reduced H&E-stained area (7.8%), which was significantly 2.4-fold lower (*p* < 0.001) compared to the H&E-stained area that emerged in the CML injected group. The treatment of DW-HDL and SO-HDL also displayed hepatic protection against CML toxicity manifested by 14.4% and 13.6% H&E-stained area, which is significantly 1.2-times (*p* < 0.05) and 1.4-times (*p* < 0.01) lesser compared to the H&E stained area quantified in the CML injected group.

The outcome of the oil red O staining, as depicted in Figure 7C,E, revealed severe fatty liver changes in the CML injected group. The 49.8% hepatic section stained with oil red O reflects the fatty liver changes observed in the CML injected group. However, treatment with SO-HDL and OSO-HDL significantly reduced this to 27% and 16.5%, respectively, and represented a remarkable 1.8-fold (*p* < 0.001) and 3.1-fold (*p* < 0.001) decrease oil red O-stained area compared to the CML injected group. Surprisingly, the DW-HDL treatment was found to be ineffective in preventing CML-induced fatty liver changes.

### 3.10. Effect of OSO on CML-Induced Hepatic Inflammation, ROS and Apoptosis

Inflammation in the hepatic tissue was quantified by measuring IL-6 levels, employing IHC staining (Figure 8A,B). A massive IL-6 production corresponding to 15.2% IHC stained area was perceived in the CML injected group (Figure 8A,B,E). Treatment of SO-HDL and OSO-HDL efficiently curtailed CML-induced IL-6 production, apparent from a marked reduction of 8.5% and 4.6% of IHC stained area that accounted for a significant 1.8-fold (*p* < 0.05) and 3.3-fold lower (*p* < 0.001) IHC stained area than appeared in the CML injected group. When compared to SO-HDL, OSO-HDL displayed a higher efficacy, manifested by a 1.8-fold reduced IHC stained area quantified in the OSO-HDL treated group than the SO-HDL injected group.

A massive ROS production, as evidenced by the highest DHE fluorescent intensity, was observed in the CML injected group, which is significantly curtailed by 1.3-fold (*p* < 0.05), 1.4-fold (*p* < 0.01), and 2.5-fold (*p* < 0.001) after the treatment of DW-HDL, SO-HDL, and OSO-HDL, respectively (Figure 8C,F). However, the best effect was exerted by the OSO-HDL as indicated by 4.2-fold and 3.8-fold reduced DHE fluorescent intensity compared to the fluorescent intensity quantified in the DW-HDL and SO-HDL injected groups, respectively.

The extent of apoptosis examined by AO fluorescent staining revealed a significantly 1.4-times (*p* < 0.001), 1.5-times (*p* < 0.001), and 5.9-times diminished apoptosis in DW-HDL, SO-HDL, and OSO-HDL injected groups, respectively, compared to CML injected group (Figure 8C,F). The highest apoptosis inhibition was observed in the OSO-HDL injected group, marked by 1.2-fold and 2.1-fold reduced AO fluorescent intensity compared to the DW-HDL and SO-HDL injected groups, respectively. The finding demonstrated a substantial preventive effect of OSO-HDL to suppress CML-triggered hepatic IL-6, ROS production, and apoptotic cell death.

### 3.11. Hepatic Function Biomarkers

Elevated plasma hepatic function biomarkers (AST and ALT) levels were quantified in the CML-injected group, and this was significantly regressed by the treatment of HDLs (Figure 9). A significant 3.1-times (*p* < 0.001), 3.3-times (*p* < 0.001), and 4.4-times (*p* < 0.001) reduced AST level was detected in the DW-HDL, SO-HDL, and OSO-HDL groups, respectively, compared to only the CML injected group. Likewise, the CML-alleviated ALT level was significantly reduced by 5.1-times (*p* < 0.001), 6.5-times (*p* < 0.001), and 6.9-times (*p* < 0.001) in the DW-HDL, SO-HDL, and OSO-HDL groups, respectively, compared to the only CML injected group. The hepatic function biomarker’s findings aligned with the hepatic histology results, validating the hepatoprotective aspect of HDLs, specifically OSO-HDL.

### 3.12. Influence of OSO-HDL on Dyslipidemia Induced CML

The plasma lipid profile in CML and CML cotreated with DW-HDL, SO-HDL, and OSO-HDL is documented in Figure 10. The highest TC level was detected in the CML injected group, which was reduced by 38.2%, 24.1%, and 30.6% by treating with DW-HDL, SO-HDL, and OSO-HDL, respectively. Surprisingly, the DW-HDL showed superiority over OSO-HDL in curtailing CML-induced TC levels. A 7.7% lower TC level was quantified in the DW-HDL group compared to the TC level observed in the OSO-HDL group (Figure 10). Consistent with the findings of TC, an elevated TG level was detected in the CML-injected group. Treatment with DW-HDL and OSO-HDL significantly reduced the TG level by ~34.2% compared to the TG level of the CML injected group (Figure 10). Interestingly, SO-HDL displayed a non-significant effect in curtailing CML-induced TG levels.

The CML-depleted HDL-C level was significantly enhanced by 32% by OSO-HDL treatment (Figure 10). Unlike OSO-HDL, the DW-HDL and SO-HDL showed a non-significant effect on the CML-alleviated HDL-C level. Also, the lowest HDL-C/TC level was detected in the CML injected group, that was significantly 2.1-times (*p* < 0.01) and 1.9-times (*p* < 0.01) elevated by treatment with DW-HDL and OSO-HDL (Figure 10). The results infer the proficiency of HDLs, precisely DW and OSO-HDL, in maintaining the lipid profile disturbed by external stress posed by CML.

## 4. Discussion

Most studies on OSO applications have focused on its topical application, with few studies examining the effect of OSO intake by animals [13,18,19]. One of our recent studies documented the impact of OSO dietary consumption for 2 years on zebrafish [18]. Additionally, in another investigation involving zebrafish, we observed the significant efficacy of OSO in mitigating dyslipidemia induced by a high-cholesterol diet [19]. Nevertheless, there is a lack of studies examining the effects of OSO consumption on plasma antioxidant activity and the functionality of lipoproteins. Considering this, the present study was performed, and findings illustrate the non-toxic effect of OSO consumption over 4 weeks on the hematological and blood biochemical profile of rats. Furthermore, the OSO-supplements group exhibited increased albumin content alongside reduced serum IL-6 and LDL-C levels compared to the SO-supplemented group. Thus, it highlights the significance of ozone-catalyzed compounds in enhancing the functionality of SO.

Likewise, a higher FRA activity was observed in the OSO-supplemented group compared to the SO and control groups, suggesting a positive impact of OSO on improving serum antioxidant status. FRA of plasma is an important marker that depicts the total antioxidant capacity of blood and reflects better defense from oxidative stress [29]. These results agree with our previous study, where we documented the in vitro FRA activity of OSO [10].

Congruent with the outcome of the FRA assay, a higher sulfhydryl content was observed in the plasma from the OSO-supplemented group, contrary to the control and SO-supplemented groups. Sulfhydryl groups are considered important antioxidants [30,31], and their diminished level has been documented in many pathological conditions like coronary artery disease and rheumatoid arthritis [32]. Protein-integrated amino acids are major free sulfhydryl groups of plasma [30] that effectively scavenge the peroxyl radicals [30], leading to a substantial antioxidant defense against plasma lipid peroxidation [30]. A combined outcome attained from the FRA and sulfhydryl content signifies a substantial modulatory effect of OSO consumption on the plasma antioxidant activity. We believe that the higher antioxidant activity of plasma from the OSO supplemented group can be attributed to the directed involvement of OSO (as an antioxidant), additionally by OSO’s positive impact on the sulfhydryl group, mainly by elevating the expression of albumin (a major plasma protein) which harbors most of the extracellular thiols [31]. So far, no reports have documented the effect of OSO consumption on plasma antioxidant status. However, one study revealed the gastro-protective effect of OSO mediated by activity enhancement of antioxidant enzymes glutathione peroxidase (GSH-Px) and superoxide dismutase (SOD) [20], signifying the positive impact on OSO on the cellular antioxidants.

The plasma obtained from the OSO-supplemented group was administered to the embryos of CML-impaired zebrafish to ascertain the enhanced plasma antioxidant status of the OSO-supplemented group compared to the plasma from the control and the SO-supplemented group. The CML, an advanced end glycation product (AGE), has been well recognized as an inducer of oxidative stress [33,34] and apoptosis [35]; consequently, it is employed to induce severe oxidative stress in the zebrafish embryos. The finding revealed the influential role of plasma from the OSO-supplemented group to alleviate the CML-induced oxidative stress and apoptosis that leads to higher embryo survivability compared to the plasma from SO and the control group and confirms the substantial impact of OSO consumption to improve the antioxidant status of plasma.

Our previous in vitro study outlined the impact of OSO on the functionality and stability of HDL [10]; however, we did not verify these effects following the consumption of OSO. To close this disparity, we now isolated lipoproteins (HDL and LDL) from the blood of OSO-supplemented rats and investigated lipid peroxidation, glycation, stability, and structural alteration. The lipid peroxidation and morphological characteristics of LDL from the OSO-supplemented group were comparable to those of LDL from the control and SO group, conveying that OSO supplementation had no detrimental effect on LDL.

The remarkable effect of the OSO supplementation was noticed in the morphology, and glycation of apoA-I/HDL. ApoA-I is the major HDL protein (70%) and substantially affects HDL’s functionality. It has been well-documented that glycation of apoA-I severely impacts the structure and functionality of apoA-I and HDL [36,37,38,39]. The HDL from the OSO group displayed the least glycation (FI glycated) and the enhanced particle diameter compared to the SO and controlled HDL, indicating the affirmative effect of OSO on HDL functionality. Of note, OSO displayed a substantial effect on the structural stability of apoA-I/HDL examined by a WMF (Ex = 295 nm, and Em = 305~400 nm) of Trp residue that is recognized as a vital approach for assessment of protein structure and dynamics [40]. ApoA-I, a major protein of HDL, harbors the Trp amino acids that are embedded inside the 3D structure and interact with the acyl radicals of phospholipids. In contrast to this, the charged amino acids in apoA-I interact with the polar residue of phospholipids in the aqueous environment [41] and stabilize the apoA-I/HDL structure. The structural modification of apoA-I/HDL resulted in the exposure of embedded Trp to the aqueous phase, consequently leading to the red shift in the Trp WMF [41], impairing HDL functionality. Therefore, a substance that can hinder the redshift of Trp WMF substantially affects the structural integrity of apoA-I/HDL. The study outcomes revealed that the Trp WMF of the apoA-I/HDL from the OSO is slightly blue-shifted compared to the Trp WMF of the apoA-I/HDL from the control and SO group. These findings strengthen the impact of OSO supplementation on the structural integrity of apoA-I/HDL. Likewise, bigger HDL particles were found to be functionally superior to smaller HDL and studies documented the altered particle size in different pathophysiological conditions [39]. The HDL from the OSO groups displayed higher particle size than the HDL particles of SO and control groups, suggesting the OSO effect on the functionality of HDL.

In addition to the particle size and glycation, the PON-1 also contributes significantly to the functionality of HDL. PON-1 is an important antioxidant enzyme present in the HDL [39,42,43]. In general, the higher PON-1 activity asserts the functional superiority of HDL [44,45]. In numerous pathological conditions like rheumatoid arthritis, impaired PON-1 activity and reduced HDL quantity and particle size were noted [39]. In the present study, we observed an enhanced PON-1 activity in HDL from the OSO-supplemented group, reflecting a substantial effect of OSO on HDL functionality. The enhanced PON-1 activity in the HDL from OSO compared to the control and SO groups might be associated with the least glycation in this group as the adverse effect of glycation is well established with the activity of PON-1 [46]. To the best of our knowledge, no study has described the impact of OSO supplementation on the structure and functionality of HDL so far; however, our earlier in vitro study documented the effect of OSO on enhancing the particle size and PON-1 activity in HDL. The current findings testify to our earlier in vitro results and strengthen the impact of OSO on the lipoprotein’s structural stability and functionality [10].

HDL has a pleiotropic function [47,48], including antioxidant [49] and anti-inflammatory [50] properties, and is well associated with cardiovascular health [51]; though, HDL involvement in various healthy and pathological conditions is also well described [52,53]. The present study outcome revealed a substantial effect of OSO consumption on enhancement of blood antioxidant ability and HDL functionality. To further validate the beneficial effects, we have examined the functionality of OSO-HDL by using embryos and adult zebrafish as model organisms of hyperinflammation. The OSO-HDL injection demonstrates a higher embryo protective effect against CML-induced toxicity than the HDL from SO and the control group, signifying the higher efficacy of OSO-HDL in rescuing embryos from CML-induced oxidative stress. We speculate the higher embryo protective activity of OSO-HDL insight due to its enhanced size and antioxidant activity that make it functionally more active to curtail the CML-induced ROS generation and consequently inhibit apoptotic cellular death of zebrafish embryos. There are no such studies documenting the HDL functionality of the zebrafish. However, reconstituted HDL (rHDL) with the enhanced HDL particle, low glycation, and high antioxidant activity deciphered the embryo’s protective effect against CML-posed toxicity [23].

Furthermore, the OSO-HDL displayed a substantial curative effect against the CML-induced paralysis by restoring the zebrafish swimming activity compared to the HDL from SO and control groups. The mechanism responsible for such an action needs to be explored. However, the most probable reason for this is the higher antioxidant and anti-inflammatory activity of OSO-HDL that curtailed CML posed oxidative stress and inflammation and restored the swinging ability. The earlier reports support the viewpoint indicating a correlation between paralysis and inflammation (explicitly linked to IL-6 production) [28]. In support of this, our previous finding highlighted a substantial impact of tocilizumab (an IL-6 inhibitor) on the recovery of CML-impaired zebrafish swimming activity, underscoring a pivotal contribution of IL-6-mediated inflammation behind the CML-induced acute paralysis. We noticed the influential role of OSO-HDL in diminishing the CML-induced IL-6 production, consequently impacting the rapid and higher recovery of zebrafish from acute paralysis.

OSO-HDL also displayed a substantial hepatoprotective role by preventing CML-induced fatty liver changes, IL-6 production, and ROS generation. The reason behind the hepatoprotective role of OSO-HDL is its substantial antioxidant effect that counters CML-induced oxidative stress and protects against hepatic damage and fatty liver changes. This notion agrees with the previous findings underscoring the impact of oxidative stress on hepatic injury and fatty liver changes [54,55]. The provocative role of oxidative stress is also well documented to induce the inflammatory pathway [56]; therefore, a reduced IL-6 production might be due to the antioxidant nature of OSO-HDL. The liver is an important site associated with lipid metabolism [57,58]; thus, better hepatic health in the OSO-HDL group is a crucial contributor to maintaining the CML-altered lipid profile. Studies documenting the involvement of proinflammatory IL-6 with disturbed TG [59,60] support the notion. Even more, many of the anti-inflammatory drugs to treat rheumatoid arthritis, psoriasis, and systemic lupus erythematous have a significant impact on the serum lipid profile [61], suggesting a deep association between inflammation and the lipid profile. We postulated that a reduced IL-6 level following OSO-HDL triggers key processes to sustain the CML-induced alterations in the lipid profile.

## 5. Conclusions

OSO supplementation over four weeks exhibited a non-toxic impact on the rat’s hematological, biochemical, and lipid profiles. Distinct from SO and the control groups, OSO supplementation manifested significantly better plasma antioxidant activity and higher sulfhydryl content. Furthermore, OSO supplementation ameliorated HDL size and PON-1 activity, along with reduced glycation within the HDL particles. The OSO-HDL logged for much better functionality to rescue zebrafish embryos and adults from CML-evoked toxicity than the HDL from SO and control groups. Additionally, OSO-HDL averted CML-induced hepatic damage, fatty liver changes, oxidative stress, and IL-6 production and maintained the blood lipid profile in zebrafish (Figure 11). The findings affirm the safe nature of OSO for enhancing plasma antioxidant status and improving HDL functionality, which can serve as a nutraceutical for combating inflammation and dyslipidemia-related disorders.

## Figures and Tables

**Figure 1 antioxidants-13-00529-f001:**
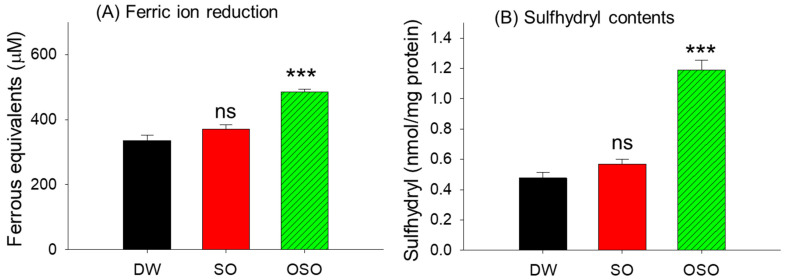
Comparative antioxidant activity of plasma obtained from the 4-week consumption of sunflower oil (SO) and ozonated sunflower oil (OSO). (**A**) Ferric ion reduction assay (FRA). (**B**) Quantification of the sulfhydryl content. *** represents *p* < 0.001 compared to the DW-supplemented group, employing one-way ANOVA with Dunnett’s post hoc test; ns denotes a non-significant difference among the groups. DW represents the distilled water-supplemented group; SO and OSO represent sunflower oil and ozonated sunflower oil-supplemented groups, respectively.

**Figure 2 antioxidants-13-00529-f002:**
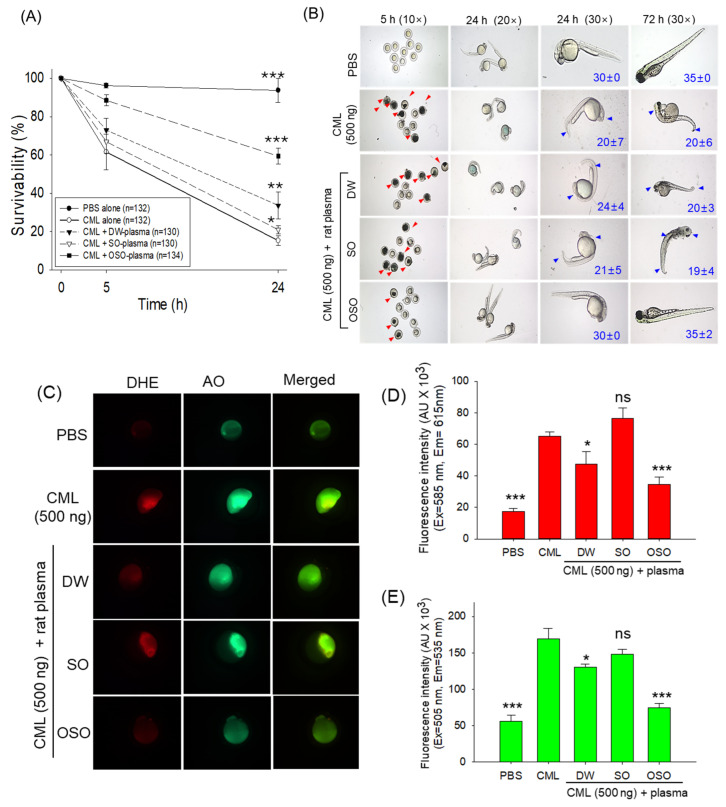
Investigating the impact of plasma obtained from rats receiving either sunflower oil (SO) or ozonated sunflower oil (OSO) supplementation over 4 weeks on the toxicity induced by carboxymethyllysine (CML) in zebrafish embryos. (**A**) Assessment of embryo survivability within 24 h post-treatment. (**B**) Visual documentation of embryo development during 72 h post-treatment (red arrows indicate dead embryos, while blue arrows designate developmental deformities); numerical values in blue font represent the somite counts. (**C**) Dihydroethidium (DHE) and acridine orange (AO) fluorescent staining to evaluate the ROS production and apoptosis, respectively [40× magnified images]. (**D**,**E**) Image J software (version 1.53r, http://rsb.info.nih.gov/ij/ assessed on 16 January 2023)-based quantification of DHE and AO fluorescent intensity, respectively. The * represents *p* < 0.05, ** represents *p* < 0.01, and *** represents *p* < 0.001 compared to the only CML-supplemented group, employing one-way ANOVA following Dunnett’s post hoc test. The PBS group were administered a microinjection containing only PBS, while the CML group received a microinjection containing solely CML (500 ng), while DW, SO, and OSO groups received microinjections of 500 ng CML+ plasma derived from distilled water (DW), sunflower oil (SO), and ozonated sunflower (OSO) supplemented groups, respectively.

**Figure 3 antioxidants-13-00529-f003:**
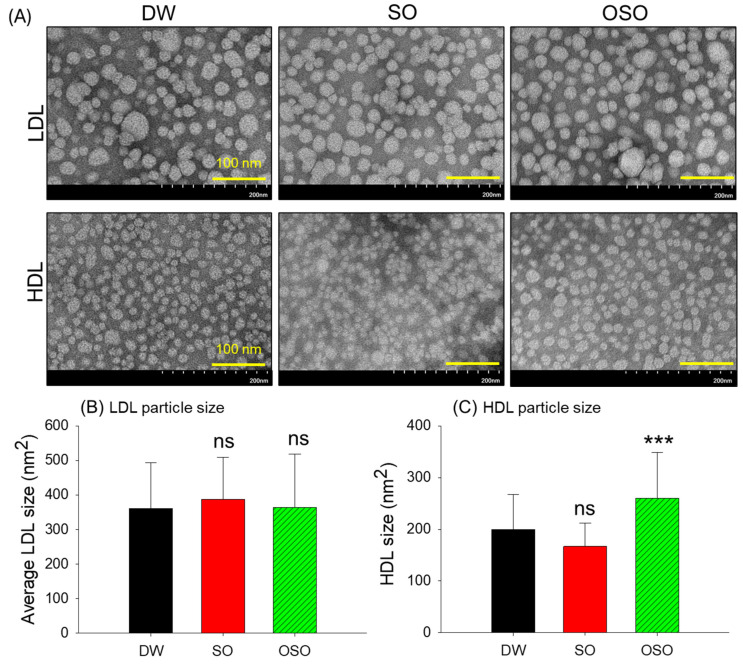
Images obtained through transmission electron microscopy (TEM) depicting the morphology of low-density lipoproteins (LDL) and high-density lipoproteins (HDL) isolated from 4-weeks sunflower oil (SO) and ozonated sunflower oil (OSO) supplemented rats. (**A**) LDL and HDL morphology in different groups. Images were taken at a magnification of 150 k following negative staining with phosphotungstic acid [scale, 100 nm]. (**B**,**C**) Average particle size of LDL and HDL, respectively, in different groups. *** represents *p* < 0.001 compared to the DW-supplemented group, employing one-way ANOVA following Dunnett’s post hoc test; ns denotes a non-significant difference among the groups. DW represents the distilled water-supplemented group; SO and OSO represent sunflower oil and ozonated sunflower oil-supplemented groups, respectively.

**Figure 4 antioxidants-13-00529-f004:**
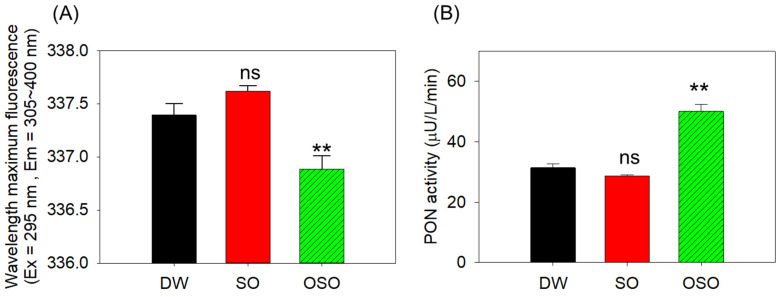
Glycation extent of apolipoprotein A-I (apoA-I) and functionality of high-density lipoprotein (HDL) isolated from the blood of rats supplemented with sunflower oil (SO) and ozonated sunflower oil (OSO) for 4 weeks. (**A**) Examination of wavelength maximum fluorescence (WMF) of tryptophan emission spectrum (excitation at 295 nm and emission at 305–400 nm). (**B**) Paraoxonase (PON)-1 activity in the HDL. ** represents *p* < 0.01 compared to the DW-supplemented group, employing one-way ANOVA following Dunnett’s post hoc test; ns denotes a non-significant difference among the groups. DW represents the distilled water-supplemented group; SO and OSO represent sunflower and ozonated sunflower oil-supplemented groups, respectively.

**Figure 5 antioxidants-13-00529-f005:**
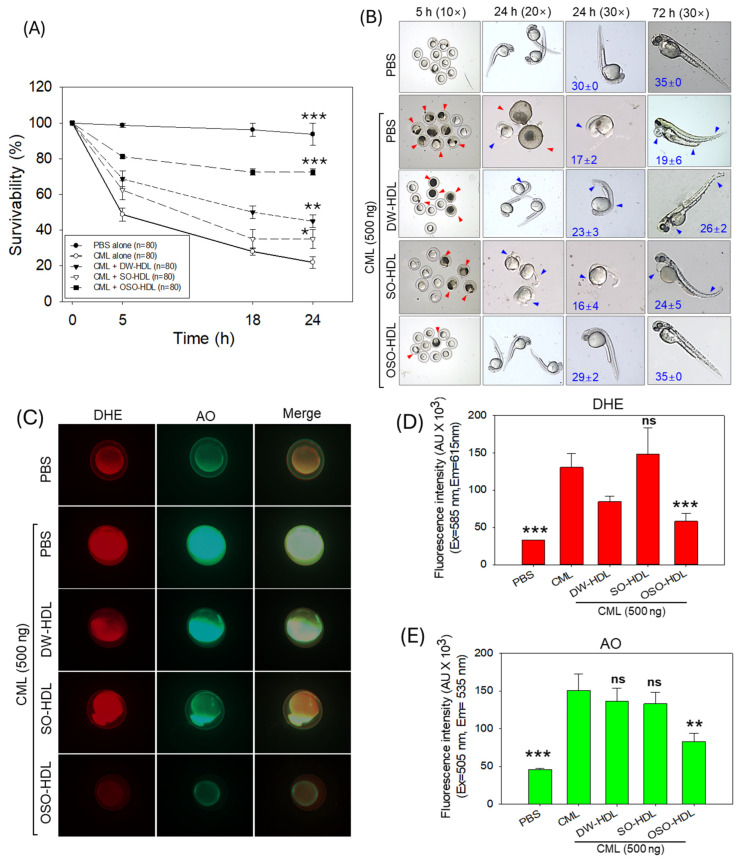
The effect of the high-density lipoprotein (HDL) isolated from the blood of rats supplemented with sunflower oil (SO) and ozonated sunflower oil (OSO) for 4 weeks on the toxicity induced by carboxymethyllysine (CML) in zebrafish embryos. (**A**) Assessment of embryo survivability within 24 h post-treatment. (**B**) Visual documentation of embryo development during 72 h post-treatment (red arrows indicate dead embryos, while blue arrows indicate developmental deformities); numerical values in blue font represent somite counts. (**C**) Dihydroethidium (DHE) and acridine orange (AO) fluorescent staining to evaluate the ROS production and apoptosis, respectively [40× magnified images]. (**D**,**E**) Image J software (version 1.53r, http://rsb.info.nih.gov/ij/ assessed on 16 January 2023)-based quantification of DHE and AO fluorescent intensity, respectively. * represents *p* < 0.05, ** represents *p* < 0.01, and *** represents *p* < 0.001 compared to the only CML-supplemented group, employing one-way ANOVA following Dunnett’s post hoc analysis. The PBS group was administered a microinjection containing only PBS, while the CML group received the microinjection containing solely CML (500 ng); and DW-HDL, SO-HDL, and OSO-HDL groups received microinjections of 500 ng CML+ HDL isolated from the blood of rats supplemented with the distilled water (DW); sunflower oil (SO), and ozonated sunflower (OSO), respectively.

**Figure 6 antioxidants-13-00529-f006:**
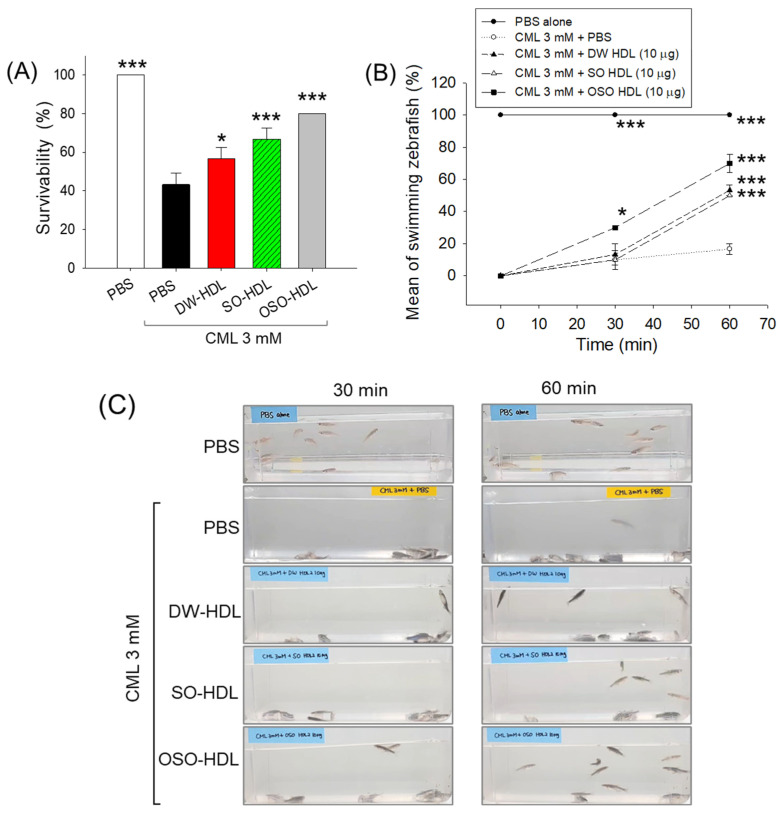
The effect of the high-density lipoprotein (HDL) isolated from the blood of rats supplemented with sunflower oil (SO) and ozonated sunflower oil (OSO) for 4 weeks on the carboxymethyllysine (CML) impaired survivability and swimming activity of adult zebrafish. (**A**) Survivability of zebrafish at 3 h post-treatment of CML alone or with HDL isolated from different groups. (**B**,**C**) Zebrafish swimming activity at 30- and 60 min post-treatment. The * represents *p* < 0.05, and *** represents *p* < 0.001 compared to the only CML-supplemented group, employing one-way ANOVA following Dunnett’s post hoc test. The PBS group was administered a microinjection containing only PBS, while the PBS + CML group received microinjections containing solely CML (3 mM) dissolved in PBS; while the DW-HDL, SO-HDL, and OSO-HDL groups received microinjections of 3 mM CML+ HDL extracted from the blood of rats supplemented with distilled water (DW), sunflower oil (SO), and ozonated sunflower oil (OSO), respectively.

**Figure 7 antioxidants-13-00529-f007:**
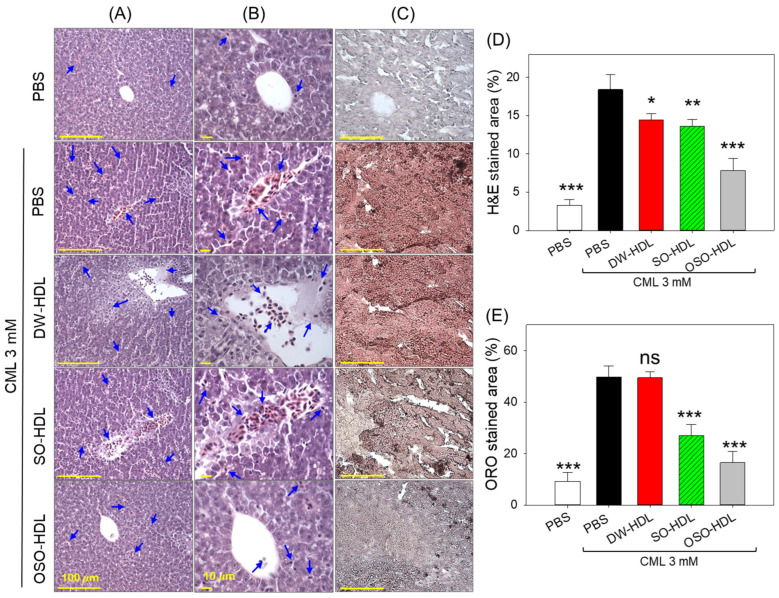
A comparative effect of the high-density lipoprotein (HDL) isolated from the blood of rats supplemented with sunflower oil (SO) or ozonated sunflower oil (OSO) for 4 weeks on the hepatic histology of carboxymethyllysine (CML) injected zebrafish. (**A**,**B**) Hematoxylin and Eosin (H&E) staining of the liver at 400× and 1000× magnification, respectively (Scale bar is 100 μm and 10 μm for the images captured at 400× and 1000× magnification, respectively]. Blue arrows indicate neutrophils. (**C**) Oil red O staining. (**D**,**E**) Quantification of H&E stained and oil red O-stained areas, respectively, utilizing Image J software (version 1.53r, http://rsb.info.nih.gov/ij/ assessed on 16 January 2023). * represents *p* < 0.05, ** represents *p* < 0.01, and *** represents *p* < 0.001 compared to the only CML-injected group, employing one-way ANOVA following Dunnett’s post hoc test. The PBS group was administered a microinjection containing only PBS, while the PBS + CML group received a microinjection containing solely CML (3 mM) dissolved in PBS; and the DW-HDL, SO-HDL, and OSO-HDL groups received microinjections of 3 mM CML+ HDL extracted from the blood of rats supplemented with distilled water (DW); sunflower oil (SO), and ozonated sunflower oil (OSO), respectively.

**Figure 8 antioxidants-13-00529-f008:**
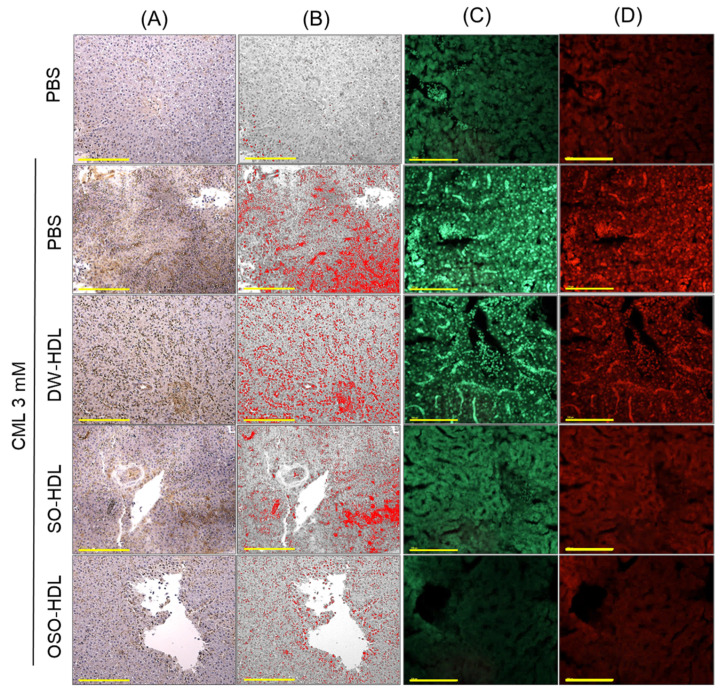
A comparative effect of the high-density lipoprotein (HDL) isolated from the blood of rats supplemented with sunflower oil (SO) and ozonated sunflower oil (OSO) for 4 weeks on inflammation, ROS, and apoptosis in the hepatic tissues of carboxymethyllysine (CML) injected zebrafish. (**A**) Interleukin (IL)-6 generation determined by immunocytochemistry (IHC) staining. (**B**) IL-6-stained area converted to red color at the brown color threshold value 20–120 using Image J software. (**C**,**D**) Dihydroethidium (DHE) and acridine orange (AO) fluorescent staining for the detection of ROS and apoptosis, respectively. Scale bar is 100 μm. (**E**,**F**) Quantification of IL-6-stained areas and DHE, AO fluorescent stained intensity, respectively, employing Image J software (version 1.53r, http://rsb.info.nih.gov/ij/ assessed on 16 January 2023). The * represents *p* < 0.05, ** represents *p* < 0.01, and *** represents *p* < 0.001 compared to the only CML-injected group for IL-6-stained area and AO fluorescent intensity; while ^##^ represents *p* < 0.01, and ^###^ represents *p* < 0.001 compared to the only CML-injected group for DHE fluorescent intensity. Dunnett’s post hoc test. The PBS group was administered a microinjection containing only PBS, while the PBS + CML group received a microinjection containing solely CML (3 mM) dissolved in PBS; and the DW-HDL, SO-HDL, and OSO-HDL groups received microinjections of 3 mM CML+ HDL extracted from the blood of rats supplemented with distilled water (DW); sunflower oil (SO), and ozonated sunflower oil (OSO), respectively.

**Figure 9 antioxidants-13-00529-f009:**
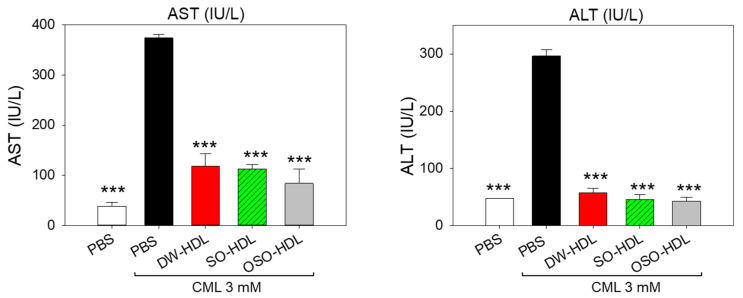
A comparative effect of the high-density lipoprotein (HDL) isolated from the blood of rats supplemented with sunflower oil (SO) and ozonated sunflower oil (OSO) for 4 weeks on the hepatic function biomarkers [aspartate aminotransferase (AST) and alanine aminotransferase (ALT). *** represents *p* < 0.001, employing one-way ANOVA following Dunnett’s post hoc test. The PBS group administered a microinjection containing only PBS, while the PBS + CML group received a microinjection containing solely CML (3 mM) dissolved in PBS; and the DW-HDL, SO-HDL, and OSO-HDL groups received microinjections of 3 m CML+ HDL extracted from the blood of rats supplemented with distilled water (DW); sunflower oil (SO), and ozonated sunflower oil (OSO), respectively.

**Figure 10 antioxidants-13-00529-f010:**
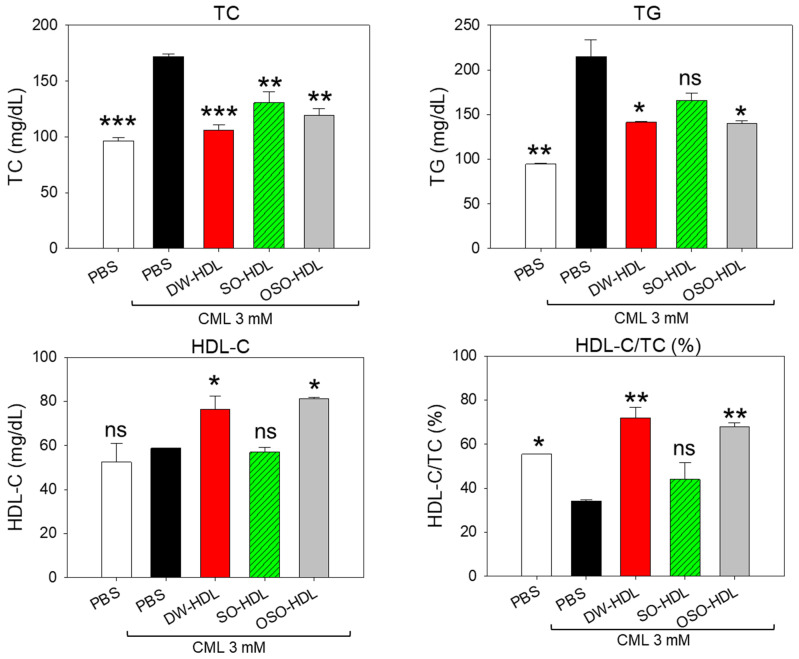
A comparative effect of the high-density lipoprotein (HDL) isolated from the blood of rats supplemented with sunflower oil (SO) and ozonated sunflower oil (OSO) for 4-week on the blood lipid profile of carboxymethyllysine (CML) injected zebrafish. The * represents *p* < 0.05, ** represents *p* < 0.01, and *** represents *p* < 0.001 compared to the only CML-injected group, employing one-way ANOVA following Dunnett’s post hoc test. TC, TG, HDL-C are the abbreviations for total cholesterol, triglycerides, and high-density lipoproteins. The PBS group was administered a microinjection containing only PBS, while the PBS + CML group received a microinjection containing solely CML (3 mM) dissolved in PBS; and the DW-HDL, SO-HDL, and OSO-HDL groups received microinjections of 3 mM CML + HDL extracted from the blood of rats supplemented with distilled water (DW); sunflower oil (SO), and ozonated sunflower oil (OSO), respectively.

**Figure 11 antioxidants-13-00529-f011:**
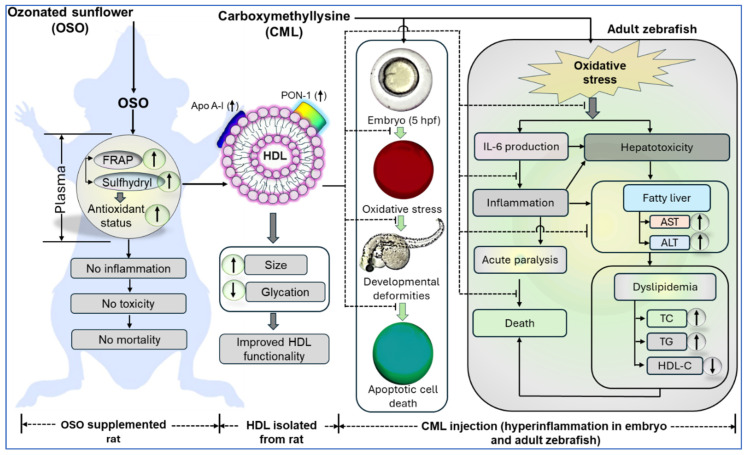
Effect of ozonated sunflower oil (OSO) supplementation on the enhancement of the antioxidant status of rat plasma, high-density lipoprotein (HDL) size and functionality, and HDL-induced events to prevent carboxymethyllysine (CML) induced toxicity in adult and embryos of zebrafish. IL-6 represents interleukin 6; AST, aspartate transferase; ALT, alanine aminotransferase; TC total cholesterol; and TG triglycerides.

**Table 1 antioxidants-13-00529-t001:** Experimental design, survivability, and changes in the body weight of male rats during 4-week consumption of sunflower oil (SO) and ozonated sunflower oil (OSO).

	Group 1	Group 2	Group 3
Oral supplementation	DW	SO	OSO
Intake amount (g/kg BW/day)	3.0	3.0	3.0
Number of rats (week 0)	(*n* = 10)	(*n* = 10)	(*n* = 10)
Number of rats (week 4)	(*n* = 10)	(*n* = 10)	(*n* = 10)
Body weight (BW) week 0 (g)	153.1 ± 6.3 ^ns,1)^	153.8 ± 6.5 ^ns,1)^	153.2 ± 6.4 ^ns,1)^
Body weight (BW) week 4 (g)	299.4 ± 22.0 ^ns,1),^ ***^,2)^	300.4 ± 15.6 ^ns,1),^ ***^,2)^	297.2 ± 13.5 ^ns,1),^ ***^,2)^
Body weight (BW) change (g) ^3)^	146.3 ± 26.9 ^ns,1)^	146.6 ± 21.1 ^ns,1)^	144.0 ± 19.7 ^ns,1)^

Data are presented as the average ± SD (standard deviation). ^1)^ indicates statistical difference in a row using Tukey’s post hoc analysis; ns represents non-significant difference between the groups. ^2)^ indicates statistical difference in the column; *** represents *p* < 0.001 compared to the BW week 0 group, with the BW of week 4, in the respective groups. ^3)^ BW change calculated by subtracting BW at 4 weeks with BW at 0 week in the respective groups. DW represents distilled water; SO and OSO represent sunflower oil and ozonated sunflower oil, respectively.

**Table 2 antioxidants-13-00529-t002:** Hematological profile following a 4-week sunflower oil (SO) and ozonated sunflower oil (OSO) supplementation.

Parameters	Group 1 DW (*n* = 10)	Group 2 SO (*n* = 10)	Group 3 OSO (*n* = 10)	Statistical Difference (*p* Value)		
				DW vs. SO	SO vs. OSO	DW vs. OSO
WBC (10^3^/μL)	8.1 ± 1.8	6.9 ± 1.8	8.7 ± 2.5	0.431	0.155	0.791
RBC (10^6^/μL)	7.5 ± 0.2	7.6 ± 0.2	7.2 ± 0.3	0.914	0.006	0.017
Hematocrit (%)	44.2 ± 1.5	44.6 ± 1.1	42.8 ± 1.9	0.831	0.034	0.113
Platelet (10^3^/μL)	1045 ± 82	1099 ± 113	1108 ± 106	0.468	0.978	0.359
Neutrophil (%)	9.8 ± 2.1	10.9 ± 3.1	10.7 ± 2.6	0.620	0.994	0.683
Lymphocyte (%)	83.6 ± 3.1	83.2 ± 3.4	83.5 ± 2.3	0.949	0.961	0.999
Monocyte (%)	5.2 ± 1.5	4.4 ± 0.9	4.6 ± 1.3	0.342	0.934	0.538
Eosinophil (%)	0.7 ± 0.2	0.9 ± 0.5	0.5 ± 0.1	0.322	0.009	0.202
Large unstained cells (%)	0.6 ± 0.1	0.5 ± 0.1	0.5 ± 0.1	0.450	1.000	0.450
Basophil (%)	0.2 ± 0.1	0.1 ± 0.1	0.1 ± 0.1	0.936	1.000	0.936

Data are presented as the average ± SD (standard deviation). The *p*-value signifies the results of a one-way ANOVA with Tukey’s post hoc analysis, aiming to verify the statistical distinction among the groups. WBC, white blood cell; RBC, red blood cell. DW represents distilled water; SO and OSO represent sunflower oil and ozonated sunflower oil, respectively.

**Table 3 antioxidants-13-00529-t003:** Biochemical and lipid profile of blood following a 4-week sunflower oil (SO) and ozonated sunflower oil (OSO) supplementation.

Parameters	Reference Range	Group 1 DW (*n* = 10)	Group 2 SO (*n* = 10)	Group 3 OSO (*n* = 10)	Statistical Difference (*p* Value)		
					DW vs. SO	SO vs. OSO	DW vs. OSO
Albumin (g/dL)	2.9–4.8	3.50 ± 0.08	3.49 ± 0.08	3.64 ± 0.12	0.973	0.012	0.007
ALT (U/L)	6–114	69.25 ± 15.7	68.38 ± 7.9	58.18 ± 4.1	0.998	0.740	0.774
AST (U/L)	37–205	91.21 ± 2.9	86.91 ± 8	76.08 ± 2.5	0.826	0.146	0.335
Creatinine (mg/dL)	0.4–1.5	0.57 ± 0.03	0.57 ± 0.02	0.58 ± 0.04	1.000	0.565	0.565
CRP (mg/L)		2.8 ± 0.1	2.8 ± 0.2	2.7 ± 0.2	0.890	0.890	0.631
γ-GTP (U/L)	0.5–5.3	3.1 ± 0.4	3.1 ± 0.5	3.1 ± 0.5	0.979	0.898	0.967
Glucose (mg/dL)	44–131	109.3 ± 6.0	100.2 ± 7.0	102.3 ± 13.7	0.104	0.877	0.250
Interleukin-6 (ng/mL)	-	0.020 ± 0.001	0.024 ± 0.004	0.016 ± 0.001	0.418	0.044	0.414
TC (mg/dL)	40–281	100.7 ± 10.2	96.3 ± 1.8	83.7 ± 2.3	0.873	0.370	0.201
TG (mg/dL)	30–409	42.0 ± 4.0	42.3 ± 1.5	40.3 ± 4.4	0.998	0.918	0.942
HDL-C (mg/dL)		61.3 ± 7.9	55.3 ± 3.2	53.0 ± 2.6	0.702	0.945	0.523
LDL-C (mg/dL)		19.0 ± 2.0	20.0 ± 1.0	12.7 ± 0.3	0.854	0.017	0.032
% HDL (HDL/TC)		60.6 ± 1.7	57.4 ± 2.4	63.3 ± 1.4	0.485	0.142	0.592
TG/HDL		0.7 ± 0.0	0.8 ± 0.0	0.8 ± 0.1	0.483	1.000	0.483
LDL/HDL		0.3 ± 0.0	0.4 ± 0.0	0.2 ± 0.0	0.269	0.031	0.269
RC (mg/dL)		20.3 ± 0.7	21.0 ± 2.0	18.0 ± 0.6	0.927	0.287	0.442

Data are presented as the average ± SD (standard deviation). The *p*-value signifies the results of a one-way ANOVA with Tukey’s post hoc analysis, aiming to verify the statistical distinction among the groups. AST, aspartate transferase; ALT, alanine aminotransferase; CRP, C-reactive protein; γ-GTP, gamma-glutamyl transpeptidase; TG, triglyceride; TC, total cholesterol; HDL-C, high-density lipoprotein; LDL-C, low-density lipoprotein; RC, remnant cholesterol. DW represents distilled water; SO and OSO represent sunflower oil and ozonated sunflower oil, respectively.

**Table 4 antioxidants-13-00529-t004:** Comparative lipid composition oxidized and glycated extent in the low-density lipoprotein (LDL) and high-density lipoprotein (HDL) isolated from the blood of 4-weeks sunflower oil (SO) and ozonated sunflower oil (OSO) supplemented rats.

		Group 1 DW (*n* = 10)	Group 2 SO (*n* = 10)	Group 3 OSO (*n* = 10)	Statistical Difference (*p* Value)		
					DW vs. SO	SO vs. OSO	DW vs. OSO
LDL	TC (mg/mL)	75.9 ± 4.8	84.5 ± 3.5	87.2 ± 4.3	0.300	0.760	0.146
	TG (mg/mL)	23.9 ± 2.8	23.3 ± 3.6	23.3 ± 1.6	0.932	1.000	0.789
	MDA (μM)	6.1 ± 0.4	6.2 ± 1.0	5.2 ± 0.8	0.969	0.242	0.343
	FI (Glycated)	5583 ± 298	5452 ± 214	5140 ± 145	1.000	0.238	0.047
	Size (nm^2^)	360.1 ± 132.7	387.2 ± 121.9	363.3 ± 155.1	0.651	0.716	0.994
HDL	TC (mg/mL)	107.4 ± 3.2	98.3 ± 6.5	99.0 ± 3.9	0.442	0.952	0.067
	TG (mg/mL)	4.0 ± 0.2	4.1 ± 0.2	3.5 ± 0.2	0.724	0.157	0.314
	MDA (μM)	4.1 ± 1.0	4.8 ± 0.4	3.2 ± 0.4	0.548	0.018	0.434
	FI (Glycated)	1941 ± 73	1832 ± 41	1752 ± 64	0.098	0.318	0.002
	Size (nm^2^)	199.6 ± 67.1	166.9 ± 44.7	260.1 ± 88.7	0.092	<0.001	<0.001

Data are presented as the average ± SD (standard deviation). The *p*-value signifies the results of a one-way ANOVA with Tukey’s post hoc analysis, aiming to verify the statistical distinction among the groups. LDL, low-density lipoprotein; HDL, high-density lipoprotein; TC, total cholesterol; TG, triglyceride; MDA, malondialdehyde; FI, fluorescence intensity (Ex = 370 nm, Em = 440 nm). DW represents the distilled water-supplemented group; SO and OSO represent sunflower oil and ozonated sunflower oil-supplemented groups, respectively.

## Data Availability

The data used to support the findings of this study are available from the corresponding author upon reasonable request.

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
