# Peer review of "Oral Supplementation of Ozonated Sunflower Oil Augments Plasma Antioxidant and Anti-Inflammatory Abilities with Enhancement of High-Density Lipoproteins Functionality in Rats"

_antioxidants, 2024, doi:10.3390/antiox13050529_

Round 1

Reviewer 1 Report

The work by Cho et al. describes the effect of OSO on various factors involved in inflammation and lipoprotein metabolism in rats. This study is largely descriptive and contains a large amount of data. Although the study contains some interesting points, there are some important concerns that should be addressed, specifically regarding statistical analysis of the data. For all other comments please see detail comments.

ABSTRACT:

Abstract is too long. Please summarize your main findings and make a short conclusion.

Line 19: please use one of the words instead of both - nonetheless or even more Line 81: please delete "using a vertebrate"

line 130: please delete the extra space between pooled bloodline line 136: please delete the extra space between . and protein line 156: please use "at" instead of "of"

line 189: the title should consist of the point of the method the authors performed and not the procedure.

line 200: as above, please change the name of the title so that the reader knows what was determined, e.g. determination of ROS in zebrafish embryos, and not the method.

lines 214, 223: Please change the titles according to the points mentioned above.

line 96: By which route was SO/OSO administered? The authors state that the rats were given the treatment orally, i.e. by gavage or by some other route? This should be clearly stated. Are the authors sure that the rats were treated with the exact amount of treatment? Please explain this in more detail.

RESULTS

General: Data should be presented as mean +/- SD and not SE. There are several reasons why the presentation of SEM is not entirely correct. One of them is that SEM represents the variability of the sample mean and not the variability of the individual data points. Therefore, it can be misleading when interpreting the spread or dispersion of the data. Unlike the standard deviation, which reflects the dispersion of the individual data points around the mean, the SEM is influenced by the sample size and can give a false impression of precision.

Section 3.1. How do the authors know whether the change in body weight (about 1.9) was statistically significant or not? Please provide an appropriate statistical analysis to obtain differences in body weight between groups and the course of feeding. If there was no statistical significance, this should also be mentioned.

Line 298 (Table 3): It is not necessary to describe the exact percentages of parameters that did not reach statistical significance. Please only pay attention to significant values. Also, the authors did not specify where the differences are, e.g. IL6 was significantly reduced. Yes, this is true, but compared to which group? This should be clearly defined and stated.

Figure 1. If the authors only compared the treated groups with the control group, then the Dunnet test should be performed instead of the Tukey test. The same applies to all other figures in which this analysis was used. Please perform the Dunnet post-hoc test if the results are to be compared with the control group only, and make appropriate changes to the significances in the graphical representations if they exist and correspond to the Dunnet test.

Figure 4A. It appears that these bands represent the signal that is saturated in all samples. Furthermore, the authors have not provided an adequate loading control to which the samples are relativized. Therefore, I am not convinced that there are differences between treatments in the results presented in this way.

Minor: Match the font and font size in the figures.

DISCUSSION:

The discussion section of the manuscript is satisfactory.

Author Response

Thank you for your valuable comments and suggestions. 

Please find attached doc as point-to-point response.

Reviewer 2 Report

Consider making the title more concise, it is too long and confusing.

It is not clear why embryos and adult zebrafish were used in addition to rats.

Please revise the abstract to improve the structural and logical flow.

Please provide reference for section 2.2 for the administering dosages of SO

How does it the dosage in zebrafish correlate to rats?

Line 12 – oral consumption of OSO?

Line 13 – supplemented to rats via oral administration or topical?

Author Response

(The authors gave the same response as above.)

Round 2

Reviewer 1 Report

The authors have made substantial improvements in the work and now the paper is easier to follow. Also, my concerns have been resolved. 

no further interventions required.